# CoLLM-NAS: Collaborative Large Language Models for Efficient Knowledge-Guided Neural Architecture Search

## Abstract

The integration of Large Language Models (LLMs) with Neural Architecture Search (NAS) has introduced new possibilities for automating the design of neural architectures. However, most existing methods face critical limitations, including architectural invalidity, computational inefficiency, and inferior performance compared to traditional NAS. In this work, we present **Co**llaborative **LLM**-based **NAS** (CoLLM-NAS), a two-stage NAS framework with knowledge-guided search driven by two complementary LLMs. Specifically, we propose a Navigator LLM to guide search direction and a Generator LLM to synthesize high-quality candidates, with a dedicated Coordinator module to manage their interaction. CoLLM-NAS efficiently guides the search process by combining LLMs' inherent knowledge of structured neural architectures with progressive knowledge from iterative feedback and historical trajectory. Experimental results on ImageNet and NAS-Bench-201 show that CoLLM-NAS surpasses existing NAS methods and conventional search algorithms, achieving new state-of-the-art results. Furthermore, CoLLM-NAS consistently enhances the performance and efficiency of various two-stage NAS methods (*e.g.,* OFA, SPOS, and AutoFormer) across diverse search spaces (*e.g.,* MobileNet, ShuffleNet, and AutoFormer), demonstrating its excellent generalization.

## 1 Introduction

Designing neural architectures remains a critical task in the era of deep learning. Neural Architecture Search (NAS), a cornerstone of AutoML, is dedicated to automating the discovery of optimal neural architectures. Early NAS methods typically employ reinforcement learning (Zoph & Le, 2016; Zoph et al., 2018; Tan et al., 2019) or evolutionary algorithms (Real et al., 2017; 2019) to search for high-performing architectures. While effective, these methods require training numerous independent architectures from scratch, resulting in prohibitive computational costs. To address this issue, one-shot NAS (Liu et al., 2019; Guo et al., 2020) was developed, utilizing a weight-sharing supernet to amortize training expenses. Two-stage NAS (Guo et al., 2020; Cai et al., 2020; Chen et al., 2021) further decouples this process into two phases, *i.e.,* supernet training and architecture search. Although this approach avoids costly retraining by inheriting supernet weights, conventional search algorithms (*e.g.,* evolutionary algorithms) still require sampling and evaluating thousands of candidates in vast search spaces during the second phase to discover optimal architectures, incurring notable test costs and risking convergence to local optima.

Recently, large language models (LLMs) have emerged as novel participants in the NAS landscape, leveraging their powerful representation capabilities, code generation proficiency, and reasoning abilities to enhance the search process from diverse perspectives (Wu et al., 2024). However, most existing LLM-based NAS methods (Chen et al., 2023; Nasir et al., 2024) directly modify architectures at the code level within unconstrained programming-language token spaces. While this maximizes search flexibility, it often results in architectural invalidity, limited robustness, and independent training of each candidate. Consequently, these methods fail to surpass established NAS baselines on standard benchmarks (Deng et al., 2009; Dong & Yang, 2020) while generating substantial computational overhead (Nasir et al., 2024). To address these limitations, we propose integrating the advantages of LLMs with mature two-stage NAS methods to provide more effective and

efficient neural architecture design. Specifically, we aim to enhance the search phase of two-stage NAS by replacing conventional search algorithms with knowledge-guided LLM reasoning, enabling intelligent navigation of the search space to accelerate convergence to high-performing regions.

To this end, this paper introduces a novel **Co**llaborative **LLM**-based **NAS** framework (CoLLM-NAS) that leverages synergistic interactions between two complementary LLMs to efficiently discover high-performing architectures for two-stage NAS. Concretely, we design three key components: (1) A stateful Navigator LLM that provides search strategies through dynamic refinement, leveraging iterative evaluation feedback and historical trajectory analysis. (2) A stateless Generator LLM that synthesizes high-quality architectures according to these strategies. (3) A Coordinator module that manages inter-LLM communication, validates legality, evaluates performance, and maintains an architecture archive. At its core, CoLLM-NAS employs a collaborative framework where the Navigator and Generator LLMs work in concert, guiding the search with dual knowledge sources, *i.e.,* LLMs' inherent architectural prior and progressive insights learned from iterative feedback and historical trajectory. The stateful-stateless design further enhances the exploration-exploitation balance. Extensive experimental results demonstrate that our method can be applied to various two-stage NAS methods across diverse search spaces, consistently outperforming baselines under different resource constraints while significantly reducing search costs.

The main contributions of this work are as follows:

- We present a novel collaborative LLM-based NAS framework, CoLLM-NAS, to enhance two-stage NAS with knowledge-guided search. To our knowledge, this is the first work integrating LLMs and two-stage NAS.
- We propose three key components: a Navigator LLM to provide adaptive search strategies, a Generator LLM to synthesize high-quality architectures, and a Coordinator module to orchestrate LLMs' interaction and workflow management.
- Experiments demonstrate that our approach enhances various two-stage NAS methods across diverse search spaces both in performance and efficiency, achieving new SOTA results on ImageNet while outperforming existing LLM-based NAS methods and conventional search algorithms on NAS-Bench-201.

## 2 RELATED WORK

### 2.1 TWO-STAGE NAS

Two-stage NAS, a prominent branch of one-shot NAS, addresses the computational inefficiency of traditional NAS by employing weight-sharing mechanisms. This approach decomposes the NAS problem into two sequential phases:

$$
\begin{aligned}
w_{\mathcal{A}}^* &= \arg\min_{w_{\mathcal{A}}} \mathbb{E}_{\alpha \sim \Omega(\mathcal{A})} \left[ \mathcal{L}(w_{\mathcal{A}}(\alpha), \mathcal{D}^{\text{train}}) \right], \\
\alpha^* &= \arg\max_{\alpha \in \mathcal{A}} \mathcal{P}(w_{\mathcal{A}}^*(\alpha), \mathcal{D}^{\text{val}}) \quad \text{s.t.} \quad \text{Cost}(\alpha) \leq \Lambda,
\end{aligned}
\tag{1}
$$

where $\mathcal{A}$ denotes the search space, $w_{\mathcal{A}}$ represents shared supernet weights, $w_{\mathcal{A}}(\alpha)$ indicates subnet weights inherited from the supernet, $\Omega(\mathcal{A})$ represents the sampling strategy, $\mathcal{P}$ and $\Lambda$ are the performance evaluation function and the resource constraint, respectively. By decoupling architecture parameter optimization from network weight training, it eliminates conflicts in simultaneous updates and reduces mutual interference.

In the first stage, a weight-sharing supernet is trained via diverse sampling strategies. SPOS (Guo et al., 2020) pioneers uniform single-path sampling. FairNAS (Chu et al., 2021b) ensures operator fairness through balanced sampling. OFA (Cai et al., 2020) enables multi-scale subnet extraction via progressive shrinking, while AutoFormer (Chen et al., 2021) adapts this for vision transformers.

In the second stage, optimal architectures are searched under resource constraints. During evaluation, candidates can directly inherit weights from the trained supernet, enabling efficient performance estimation without retraining. Since exhaustive evaluation remains infeasible, most methods employ random search (Li & Talwalkar, 2020), reinforcement learning (Zoph & Le, 2016), or evolutionary algorithms (Guo et al., 2020; Cai et al., 2020) to efficiently explore the search space.

This paper introduces an LLM-based knowledge-guided search paradigm to replace conventional search algorithms in the second stage, aiming to achieve SOTA performance with lower costs.

## 2.2 LLM FOR NAS

The rise of LLMs offers a fundamentally new perspective for NAS by reformulating the search process through advanced reasoning capabilities and semantic understanding. GENIUS (Zheng et al., 2023) pioneers GPT-4 as a black-box optimizer, directly generating and iteratively refining neural architectures via natural language prompts and performance feedback. EvoPrompting (Chen et al., 2023) repositions LLMs as adaptive mutation and crossover operators, employing evolutionary prompt engineering coupled with soft prompt-tuning to explore a vast code-based search space. LLMatic (Nasir et al., 2024) similarly leverages LLMs as mutation and crossover operators, but uniquely integrates Quality-Diversity (QD) Optimization to create diverse and high-performing networks. RZ-NAS (Ji et al., 2025) introduces a reflective zero-cost strategy by combining LLM reflection modules with training-free metrics, achieving SOTA performance on NAS benchmarks. Looking forward, the rapid advancement of large reasoning models (*e.g.,* OpenAI-o3 (OpenAI, 2025), DeepSeek-R1 (Guo et al., 2025), Qwen3 (Team, 2025)) and ever-expanding pretraining corpora will promote deeper integration of LLMs into NAS methodologies.

This work proposes a collaborative LLM-based NAS framework integrated with two-stage NAS methodology. Distinct from previous LLM-based NAS methods, our approach leverages a pretrained supernet for efficient evaluation and introduces a collaboration mechanism to optimize the exploration-exploitation trade-off during search.

## 3 METHOD

In this section, we first validate LLMs' capability to comprehend structured neural architectures within hand-crafted search spaces. Based on this capability, we introduce CoLLM-NAS, our collaborative LLM-based NAS framework.

### 3.1 ARCHITECTURE COMPREHENSION IN LLMS

Recent advances in LLMs demonstrate their capability to understand complex technical domains, acquired through pre-training on extensive technical literature. *We hypothesize that modern LLMs have internalized knowledge of neural architecture design principles that could be valuable for NAS.* To investigate this capability, we design an experiment on NAS-Bench-201.

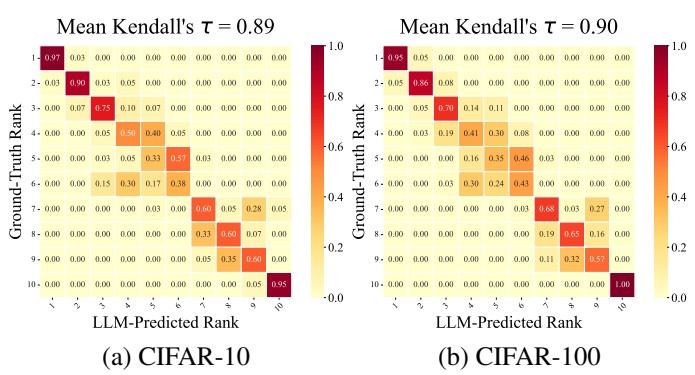

Figure 1: Consistency heatmap between LLM-predicted and ground-truth rankings on CIFAR-10 and CIFAR-100 within NAS-Bench-201 search space.

**Proof-of-Concept Experiment.** We uniformly sample 10 architectures from NAS-Bench-201 search space, each with precomputed performance metrics. The Qwen3-30B-A3B LLM (Team, 2025) is prompted to rank these architectures based on its comprehension of neural network design principles, while being blinded to ground-truth accuracy. Architectures are represented in their standard graph-based encoding format. To ensure statistical robustness, we perform 40 independent trials with varied random seeds and LLM temperature on CIFAR-10 and CIFAR-100. Ranking quality is quantified through Kendall's $\tau$, which measures ordinal correlation between predicted and ground-truth rankings. As illustrated in Figure 1, results reveal strong alignment between LLM predictions and empirical performance, achieving mean Kendall's $\tau$ values of 0.89 on CIFAR-10 and 0.90 on CIFAR-100. This demonstrates that *LLMs exhibit non-trivial comprehension of neural architecture performance patterns*, even when

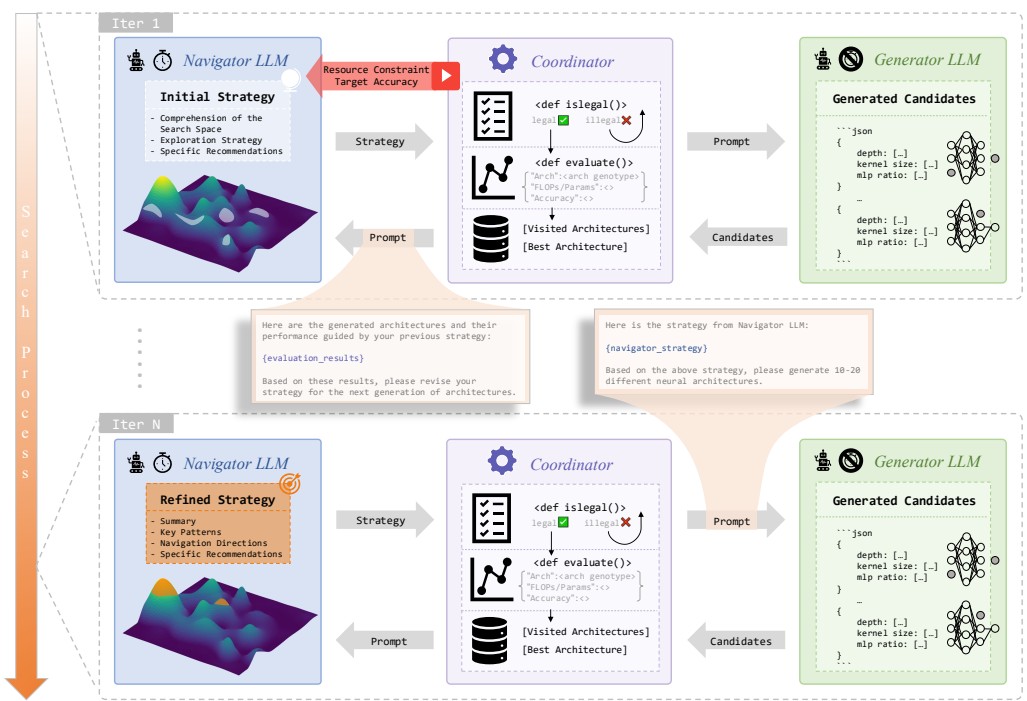

Figure 2: Pipeline of CoLLM-NAS. The search starts with the Navigator LLM generating an initial exploration strategy based on target accuracy and resource constraint (*e.g.,* FLOPs, Params). The Coordinator then transmits this strategy to the Generator LLM, which synthesizes high-quality candidates accordingly. After Coordinator validation and evaluation, results are fed back to the Navigator LLM for strategy refinement. This loop iterates until either achieving the target accuracy or reaching the iteration limit. Orange regions indicate more promising areas in the search space.

operating on structured representations within hand-crafted search spaces. Notably, the optimal architecture is correctly identified in the vast majority of trials, further validating LLMs' capability to reliably recognize top-performing architectures. Additional details appear in Appendix D.

## 3.2 CoLLM-NAS Framework

As shown in Figure 2, CoLLM-NAS consists of three key components: a Navigator LLM, a Generator LLM, and a Coordinator module that facilitates their interaction.

**Navigator LLM.** The stateful Navigator LLM functions as a strategic guide with persistent memory across iterations. Through iterative analysis of performance patterns emerging from evaluated architectures, it dynamically formulates and refines search strategies. These strategies progressively concentrate on high-potential regions of the search space. In the initial phase, the Navigator LLM is prompted to establish an exploration strategy that promotes architectural diversity, improving initial population quality through its implicit comprehension of architectures. As iterations progress, it continuously refines this strategy based on accumulated feedback, transitioning from broad exploration to targeted exploitation of identified high-performing regions.

**Generator LLM.** The stateless Generator LLM serves as a specialized architecture synthesizer, focusing exclusively on the current strategy without retaining any memory. Following Navigator LLM's guidance, it translates the strategy into concrete candidate architectures during each iteration. These candidates simultaneously conform to the search space constraints while embodying the architectural patterns emphasized by the current strategy.

**Coordinator.** The Coordinator manages the overall search process, verifying legality, evaluating performance, and handling information flow between LLMs. During each iteration, the module also maintains an archive of visited architectures to eliminate redundant evaluations and tracks the

---

**Algorithm 1** CoLLM-NAS: Exploring High-Performing Architectures via Collaborative LLMs

---

**Require:** Target accuracy $P_{target}$, Resource constraint $\Lambda$, Iteration limit $T$
**Ensure:** Best architecture $\alpha^*$
1: **Initialization:** $\alpha^* \leftarrow \emptyset, p^* \leftarrow 0, \mathcal{V} \leftarrow \emptyset, \mathcal{H} \leftarrow \emptyset$       ▷ Best arch, best accuracy, visited set, history
2:   $\mathcal{S}_0 \leftarrow \textsc{NavigatorLLM}(P_{target}, \Lambda)$       ▷ Initialize search strategy
3: **for** $t = 1$ **to** $T$ **do**
4:     $\mathcal{C}_t \leftarrow \textsc{GeneratorLLM}(\mathcal{S}_{t-1})$       ▷ Generate candidate architectures
5:     $\mathcal{R}_t \leftarrow \emptyset$       ▷ Collection of evaluation results
6:     **for** each $\alpha_i \in \mathcal{C}_t \setminus \mathcal{V}$ **such that** $\textsc{isLegal}(\alpha_i)$ **do**
7:         Compute cost $c_i$ and evaluate performance $p_i$ for $\alpha_i$
8:         $\mathcal{R}_t \leftarrow \mathcal{R}_t \cup \{(\alpha_i, p_i, c_i)\}; \mathcal{V} \leftarrow \mathcal{V} \cup \{\alpha_i\}$
9:         **if** $c_i \leq \Lambda$ and $p_i > p^*$ **then** $p^* \leftarrow p_i, \alpha^* \leftarrow \alpha_i$
10:        **end if**
11:     **end for**
12:     **if** $p^* \geq P_{target}$ **then break**
13:     **end if**
14:     $\mathcal{H} \leftarrow \mathcal{H} \cup \{(\mathcal{S}_{t-1}, \mathcal{R}_t)\}$       ▷ Update history
15:     $\mathcal{S}_t \leftarrow \textsc{NavigatorLLM}(\mathcal{H})$       ▷ Refine search strategy
16: **end for**
17: **return** $\alpha^*$

---

best-performing architecture. Critically, the evaluation employs inherited weights from a pre-trained supernet, enabling rapid performance assessment while preserving learned parameter relationships.

Through system prompt design, we assign distinct roles and responsibilities to each LLM, inform them of the collaboration process, convey knowledge about different search spaces and architectural representations, and provide guidance on output formats. Simultaneously, to avoid prior knowledge contamination, our prompts prevent the transmission of any explicit information about search spaces. Detailed examples of prompts and responses are shown in Appendix G. Our full algorithm is described in Algorithm 1. Through this collaborative approach, CoLLM-NAS intelligently navigates search spaces and efficiently identifies high-performing architectures.

# 4 EXPERIMENT

## 4.1 EXPERIMENTAL SETUP

We evaluate the proposed CoLLM-NAS in three macro search spaces: MobileNet (Cai et al., 2019), ShuffleNet (Ma et al., 2018), and AutoFormer (Chen et al., 2021), and one micro cell-based search space: NAS-Bench-201 (Dong & Yang, 2020).

**Macro Search Spaces.** In each macro search space, we adopt a two-stage NAS approach (*i.e.,* OFA (Cai et al., 2020), SPOS (Guo et al., 2020), and AutoFormer (Chen et al., 2021)) as the baseline and apply our collaborative LLM-based search to them. To ensure fair comparison, we directly reuse the pre-trained supernet from each baseline for the search phase. All baselines employ evolutionary algorithms as their search methods. Our approach implements Qwen3-30B-A3B (Team, 2025) as the foundational LLM, deployed locally via vLLM (Kwon et al., 2023), with temperature 0.6 and chain-of-thought reasoning enabled. Key experimental settings are summarized in Table 6, including evaluation methodologies, resource constraint types, and retraining requirements. Notably, OFA employs a trained accuracy predictor for rapid subnet performance estimation. However, we observe that this predictor is unreliable for distinguishing top-performing architectures (Appendix F). CoLLM-NAS therefore adopts the same evaluation methodology as SPOS and AutoFormer by directly evaluating subnets on the full validation set.

All experiments are conducted on ImageNet (Deng et al., 2009) under varying resource constraints and fixed budgets on the number of explored architectures. Search costs, covering the entire duration of the search phase (including LLM inference), are quantified in GPU days on a single NVIDIA A100-80GB GPU. For OFA, we allocate a higher architecture exploration budget but exclude search cost metrics due to the incomparable evaluation methodology. For SPOS, we note the narrow FLOPs range (280-360M) of ShuffleNet search space is unsuitable for multi-tiered constraint decomposition. We therefore preserve the original setting from SPOS (Guo et al., 2020), searching exclusively

Table 1: Performance comparison on ImageNet within macro search spaces. "GPU Days" only covers the search phase (excluding supernet training), and "Arch. Budget" refers to the budget on the number of explored architectures.

| Method | Top-1 (%) | FLOPs (M) | Params. (M) | GPU Days | Arch. Budget |
|---|---|---|---|---|---|
| MobileNet Search Space | | | | | |
| OFA-T | 75.5 | 200 | 3.2 | | |
| OFA-S | 77.2 | 299 | 4.2 | | 5000 |
| OFA-B | 78.2 | 399 | 5.7 | - | |
| OFA-L | 78.7 | 496 | 5.2 | | |
| OFA-T + Ours | 75.7 | 199 | 4.4 | | |
| OFA-S + Ours | 77.6 | 297 | 3.7 | | 250 |
| OFA-B + Ours | 78.4 | 396 | 4.5 | 0.12 | |
| OFA-L + Ours | 78.9 | 494 | 6.1 | | |
| ShuffleNet Search Space | | | | | |
| SPOS | 73.6 | 323 | 3.5 | 0.32 | 1000 |
| SPOS + Ours | 73.8 | 325 | 3.7 | 0.09 | 250 |
| AutoFormer Search Space | | | | | |
| AutoFormer-T | 74.7 | 1344 | 5.9 | | |
| AutoFormer-S | 81.6 | 4887 | 22.8 | 1.0 | 1000 |
| AutoFormer-B | 82.1 | 11305 | 54.0 | | |
| AutoFormer-T + Ours | 75.3 | 1366 | 6.0 | | |
| AutoFormer-S + Ours | 81.7 | 4897 | 22.9 | 0.1 | 250 |
| AutoFormer-B + Ours | 82.3 | 11074 | 52.8 | | |

at the 330M FLOPs constraint. Results report the best-performing subnet from three independent runs per method. More details about the search spaces are provided in Appendix C.

**Micro Search Space.** NAS-Bench-201 is a widely used cell-based search space for NAS benchmarking. It contains 15,625 architectures with precomputed performance on CIFAR-10, CIFAR-100, and ImageNet-16-120 datasets. Each architecture is represented as a directed acyclic graph (DAG) with 4 nodes and 6 edges, where every edge is assigned one of five candidate operations: *none*, *skip_connect*, *conv_1x1*, *conv_3x3*, or *avg_pool_3x3*.

We compare various approaches on NAS-Bench-201, including traditional one-shot NAS methods, emerging LLM-based NAS methods, and conventional search algorithms. To ensure fair comparison and avoid evaluation discrepancies, our method follows conventional search algorithms, *i.e.,* Random Search (RS), Reinforcement Learning (RL), and Evolutionary Algorithm (EA), in directly adopting ground-truth performance metrics as accuracy measures. All methods explore under a fixed cap of 100 architectures. Specifically, EA employs a population size of 10 over 10 iterations, with 50% elite preservation. RL operates with a learning rate of 0.01 and EMA momentum of 0.9. Our method terminates after at most 20 iterations. Results average over 10 independent runs.

## 4.2 MAIN RESULTS

**Macro Search Spaces Results.** Table 1 compares Top-1 test accuracy of subnets discovered by our method with baselines in different macro search spaces. Our method consistently outperforms baselines across all search spaces, with significantly lower search costs and fewer architecture evaluations, demonstrating its effectiveness and efficiency in discovering high-performing architectures. While our approach incurs additional overhead from LLM inference, this cost is substantially outweighed by the dramatic reduction in evaluations. Specifically, our approach achieves up to 0.6% accuracy improvements while reducing search costs by 3–10× compared to baselines. This efficiency gain is most pronounced in the AutoFormer search space, where we discover better architectures with only 10% of the baseline's search cost and 25% of the number of evaluations.

**Comparison with State-of-the-Art Methods.** We further provide a comparison with SOTA NAS methods on ImageNet in MobileNet search space. As shown in Table 2, our approach achieves a remarkable 77.9% Top-1 accuracy with only 320M FLOPs, surpassing existing SOTA methods.

Table 2: Comparison with SOTA NAS methods on ImageNet. Top: manual design. Middle: differentiable NAS methods. Bottom: two-stage NAS methods.

| Method | Top-1 (%) | Top-5 (%) | FLOPs (M) |
|---|---|---|---|
| Mobilenetv2 (Sandler et al., 2018) | 72.0 | - | 300 |
| Mobilenetv3 (Howard et al., 2019) | 76.6 | - | 350 |
| MixNet (Tan & Le, 2019) | 77.0 | 93.3 | 360 |
| DARTS (Liu et al., 2019) | 73.3 | 91.3 | 574 |
| PC-DARTS (Xu et al., 2020) | 75.8 | 92.7 | 597 |
| DARTS- (Chu et al., 2021a) | 76.2 | 93.0 | 467 |
| ProxylessNAS (Cai et al., 2019) | 74.6 | 92.2 | 320 |
| SPOS (Guo et al., 2020) | 75.6 | 92.8 | 330 |
| FairNAS (Chu et al., 2021b) | 76.7 | - | 325 |
| OFA (Cai et al., 2020) | 77.5 | 93.5 | 330 |
| GreedyNAS (You et al., 2020) | 76.8 | 93.0 | 324 |
| GreedyNASv2 (Huang et al., 2022) | 77.5 | 93.5 | 324 |
| SUMNAS (Ha et al., 2022) | 77.6 | - | 349 |
| PA&DA (Lu et al., 2023) | 77.3 | 93.5 | 399 |
| DYNAS (Jeon et al., 2025) | 75.9 | - | 329 |
| OFA + Ours | **77.9** | **93.8** | 320 |

Table 3: Test and validation accuracy on NAS-Bench-201. Top: one-shot NAS methods. Upper middle: LLM-based NAS methods. Lower middle: conventional search algorithms against our approach. [†] denotes results from (Ji et al., 2025).

| Method | CIFAR-10 | | CIFAR-100 | | ImageNet-16-120 | |
|---|---|---|---|---|---|---|
| | valid | test | valid | test | valid | test |
| DARTS | $39.77 \pm 0.00$ | $54.30 \pm 0.00$ | $15.03 \pm 0.00$ | $15.61 \pm 0.00$ | $16.43 \pm 0.00$ | $16.32 \pm 0.00$ |
| PC-DARTS | $89.96 \pm 0.15$ | $93.41 \pm 0.30$ | $67.12 \pm 0.39$ | $67.48 \pm 0.89$ | $40.83 \pm 0.08$ | $41.31 \pm 0.22$ |
| DARTS- | $91.03 \pm 0.44$ | $93.80 \pm 0.40$ | $71.36 \pm 1.51$ | $71.53 \pm 1.51$ | $44.87 \pm 1.46$ | $45.12 \pm 0.82$ |
| FairNAS | $90.07 \pm 0.57$ | $93.23 \pm 0.18$ | $70.94 \pm 0.94$ | $71.00 \pm 1.46$ | $41.90 \pm 1.00$ | $42.19 \pm 0.31$ |
| GENIUS[†] | $91.07 \pm 0.20$ | $93.79 \pm 0.09$ | $70.96 \pm 0.33$ | $70.91 \pm 0.72$ | $45.29 \pm 0.81$ | $44.96 \pm 1.02$ |
| LLMatic[†] | $91.42 \pm 0.13$ | $94.26 \pm 0.10$ | $71.41 \pm 1.44$ | $71.62 \pm 1.73$ | $44.98 \pm 0.87$ | $45.87 \pm 0.96$ |
| RZ-NAS[†] | $91.45 \pm 0.10$ | $94.24 \pm 0.12$ | $73.35 \pm 0.14$ | $73.30 \pm 0.21$ | $46.53 \pm 0.24$ | $46.24 \pm 0.23$ |
| RS | $90.96 \pm 0.24$ | $93.83 \pm 0.09$ | $71.62 \pm 0.73$ | $71.52 \pm 0.93$ | $45.52 \pm 0.48$ | $45.37 \pm 0.54$ |
| RL | $91.20 \pm 0.41$ | $93.94 \pm 0.25$ | $71.51 \pm 0.93$ | $71.78 \pm 1.20$ | $45.22 \pm 0.79$ | $45.95 \pm 0.76$ |
| EA | $91.28 \pm 0.30$ | $94.21 \pm 0.25$ | $72.17 \pm 1.13$ | $72.71 \pm 0.83$ | $46.17 \pm 0.50$ | $46.42 \pm 0.60$ |
| Ours | $\mathbf{91.59 \pm 0.04}$ | $\mathbf{94.37 \pm 0.01}$ | $\mathbf{73.44 \pm 0.12}$ | $\mathbf{73.44 \pm 0.15}$ | $\mathbf{46.62 \pm 0.10}$ | $\mathbf{46.79 \pm 0.28}$ |
| Optimal | 91.61 | 94.37 | 73.49 | 73.51 | 46.73 | 47.31 |

This demonstrates our method's enhanced capability to efficiently navigate expansive search spaces and identify superior subnets.

**Micro Search Space Results.** We present NAS-Bench-201 results in Table 3, where our approach maintains consistent superiority over other search methods across all datasets, especially achieving significant improvements over RL and EA. Our method further demonstrates enhanced robustness, evidenced by lower standard deviations. Moreover, by exploring at most 100 architectures, significantly fewer than the presented one-shot and LLM-based NAS approaches require, our method achieves SOTA performance with remarkable search efficiency.

## 4.3 DISCUSSION

Traditional two-stage NAS relies on evolutionary algorithms (EAs) that refine architectures through stochastic operators like mutation and crossover. While capable of exploration, these operators are inherently local and undirected, generating new candidates via small perturbations of elite individuals. This approach lacks a global understanding of the performance landscape, often requiring numerous evaluations to achieve significant progress, resulting in reduced search efficiency.

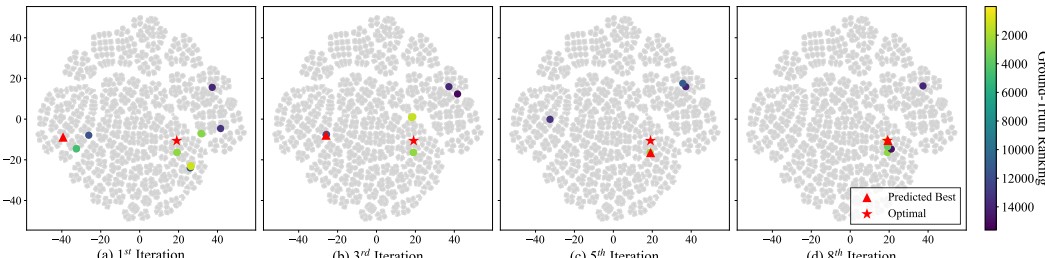

Figure 3: T-SNE visualization of CoLLM-NAS's search dynamics on ImageNet-16-120 within NAS-Bench-201 search space. Architectural performance rankings are represented through color-coding, with unexplored ones displayed in gray.

In contrast, CoLLM-NAS transforms architecture search into a directed, learning-based optimization process, aligning with the emerging "LLMs as optimizers" paradigm, where LLMs can gradually improve the generated solutions based on past optimization steps. However, compared to prior work like OPRO (Yang et al., 2024) that typically employs a single LLM to directly map optimization trajectories to solutions, our framework implements a more sophisticated, two-step generative process:

$$\mathcal{S}_t \leftarrow \text{NAVIGATORLLM}(\mathcal{H}_t), \quad \mathcal{C}_{t+1} \leftarrow \text{GENERATORLLM}(\mathcal{S}_t). \quad (2)$$

The Navigator LLM first maps the optimization trajectory $\mathcal{H}_t$ to an abstract, natural-language strategy $\mathcal{S}_t$. Then, a separate Generator LLM translates this strategy into concrete candidates $\mathcal{C}_{t+1}$. This "trajectory → strategy → solution" pipeline encourages more structured exploration by reasoning at a higher abstraction level, mitigating overfitting to specific architectural syntax and improving search robustness. We validate this hypothesis in Section 4.4.

This guided search is further enhanced by the interplay between dual knowledge sources. First, as demonstrated in Section 3.1, the LLM's inherent knowledge of effective architecture design improves the quality of generated candidates and provides a powerful "warm-start" in promising regions. Second, the progressive knowledge accumulated from optimization trajectories enables the Navigator LLM to gradually learn an implicit model of the performance landscape, and guide the Generator LLM towards increasingly promising regions. Moreover, our unique memory retention mechanism—featuring a stateful Navigator LLM paired with a stateless Generator LLM—further ensures an effective balance between exploration and exploitation. In Figure 3, we present t-SNE visualization of CoLLM-NAS's search dynamics, illustrating how the search distribution rapidly focuses on high-performance areas and efficiently locates the global optimum.

### 4.4 ABLATION STUDIES

**Main Mechanisms.** In Figure 4, we ablate our collaboration and memory retention mechanisms respectively. For collaboration, we introduce a Single LLM NAS (SiLLM-NAS) variant that maintains the reflection-then-generation paradigm but consolidates both roles into a single LLM. Over 10 runs, we track mean accuracy of the best architecture per iteration. Figure 4 (a) shows CoLLM-NAS consistently outperforms SiLLM-NAS across all datasets, verifying collaboration efficacy. Crucially, CoLLM-NAS generates better initial populations, highlighting the Navigator LLM's critical role in initial exploration.

Regarding memory retention, Figure 4 (b) shows that for low-complexity datasets (*e.g.,* CIFAR-10, CIFAR-100), optimal performance is achieved without memory retention in either LLM, indicating that iterative feedback alone suffices for simpler tasks. Conversely, for high-complexity datasets (*e.g.,* ImageNet-16-120, ImageNet), preserving the Navigator LLM's memory while disabling the Generator's yields optimal results, confirming the necessity of historical trajectory in demanding scenarios. Notably, we observe that retaining the Generator LLM's memory induces progressive noise accumulation, leading to substantial invalid architectures and performance degradation.

**Rephrasing Prompts.** To assess whether our gains depend on a particular wording, we rephrase the original prompts with three leading LLMs, *i.e.,* Claude Sonnet 4 (Anthropic, 2025), GPT-5 (OpenAI, 2025), and DeepSeek-R1 (DeepSeek-AI, 2025). As shown in Table 4, all variants achieve comparable performance across NAS-Bench-201 datasets, with Variant 2 even outperforming the

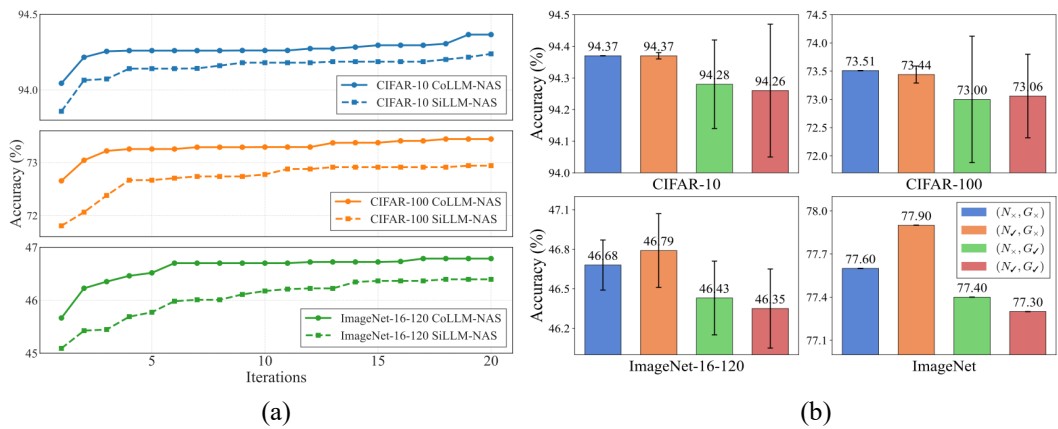

(a)  (b)

Figure 4: Ablation on main mechanisms: (a) Comparison of iterative performance between CoLLM-NAS and SiLLM-NAS on NAS-Bench-201. (b) Impact of memory retention settings on different datasets. $N_{\checkmark/\times}$ and $G_{\checkmark/\times}$ denote whether the memory of Navigator/Generator LLM is retained.

Table 4: Performance comparison with rephrasing prompts. Variant 1-3 are rephrased by Claude Sonnet 4, GPT-5, and DeepSeek-R1 respectively.

| Prompt | CIFAR-10 | CIFAR-100 | ImageNet-16-120 |
|---|---|---|---|
| Base (Ours) | **94.37±0.01** | **73.44±0.15** | 46.79±0.28 |
| Variant 1 | 94.36±0.02 | 73.35±0.52 | 46.52±0.36 |
| Variant 2 | 94.35±0.03 | 73.36±0.18 | **46.89±0.35** |
| Variant 3 | 94.16±0.23 | 73.19±0.11 | 46.60±0.29 |

Table 5: Performance comparison using different LLMs. * denotes models from (Team, 2025), and ‡ denotes models from (DeepSeek-AI, 2025).

| LLM | CIFAR-10 | CIFAR-100 | ImageNet-16-120 |
|---|---|---|---|
| Qwen3-30B-A3B* | **94.37±0.01** | **73.44±0.15** | **46.79±0.28** |
| Qwen3-32B* | 94.31±0.14 | 73.29±0.29 | 46.64±0.52 |
| DeepSeek-R1-Distill-Qwen-32B‡ | 94.36±0.04 | 73.37±0.19 | 46.53±0.32 |
| DeepSeek-R1-Distill-Llama-70B‡ | **94.37±0.00** | 73.41±0.22 | 46.74±0.31 |

original prompt on ImageNet-16-120. The narrow performance spread indicates that improvements stem from the proposed collaborative framework rather than handcrafted phrasing, evidencing robustness to linguistic reformulation.

**Different LLMs.** We further perform extensive experiments with different open-source LLMs on NAS-Bench-201. As shown in Table 5, CoLLM-NAS maintains consistently strong performance across diverse LLMs, confirming its generality beyond specific LLM implementations. Additionally, we verify CoLLM-NAS's robustness across different LLM temperature settings. Detailed experiments and results are provided in Appendix E.

## 5    CONCLUSION

In this paper, we present CoLLM-NAS, a collaborative LLM-based NAS framework that integrates LLMs with two-stage NAS. We design a stateful Navigator LLM to provide adaptive search strategies, a stateless Generator LLM to synthesize high-quality architectures, and a Coordinator module to manage LLM interaction and workflow. CoLLM-NAS employs knowledge-guided search by coupling LLMs' inherent knowledge of structured neural architectures with progressive knowledge from iterative feedback and historical trajectory. Extensive experiments demonstrate that CoLLM-NAS consistently enhances various two-stage NAS methods across diverse search spaces and outperforms existing NAS methods and conventional search algorithms, achieving new state-of-the-art results. Furthermore, the generalizable nature of CoLLM-NAS suggests promising extensions beyond NAS, revealing new avenues for future exploration.

## REPRODUCIBILITY STATEMENT

To ensure the reproducibility of our work, we have provided comprehensive details of our methodology and experiments. The core CoLLM-NAS framework is formally described in Algorithm 1. Our experimental setup, including the datasets, baselines, key experimental settings, evaluation protocols, and hardware specifications, is detailed in Section 4.1. For further clarity, we provide in-depth descriptions of the macro search spaces, in Appendix C and a sensitivity analysis of key hyperparameters, such as LLM temperature, in Appendix E. Crucially, the exact prompts and representative responses that drive our collaborative framework are documented in Appendix G, and the final discovered architectures are listed in Appendix H. To facilitate replication of our results, we have also included our core source code in the supplementary materials.

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

## A  THE USE OF LLMs IN PAPER WRITING

In paper writing, we use large language models (LLMs) only for polish writing, including improving word choice and terminology consistency, making phrasing more professional and concise, and correcting grammar and fluency. All AI-suggested edits are reviewed and accepted manually.

## B  KEY EXPERIMENTAL SETTINGS

Table 6: Key experimental settings. "*same*" indicates identical settings to the corresponding baseline.

| Method | Evaluation | Resource Constraint | Retrain |
|---|---|---|---|
| OFA | predict | FLOPs | × |
| SPOS | validate | FLOPs | ✓ |
| AutoFormer | validate | Params. | × |
| Ours | validate | *same* | *same* |

## C  MACRO SEARCH SPACES

**MobileNet Search Space.**  The MobileNet search space employs MBConv blocks as fundamental units, featuring depthwise separable convolutions and squeeze-excitation modules. Configurable dimensions include:

- *Resolution* ($r \in \{160, 176, 192, 208, 224\}$) controlling input scale,
- *Stage depth* ($d_i \in \{2, 3, 4\}$ for $i = 1$ to 5 stages) determining active blocks per stage,
- *Kernel size* ($k_j \in \{3, 5, 7\}$ for $j = 1$ to 20 convolutional layers),
- *Expansion ratio* ($e_j \in \{3, 4, 6\}$ for $j = 1$ to 20 inverted residual blocks).

The combinatorial space contains $\sim 10^{19}$ architectures, with stage-specific depth controlling block activation patterns (*e.g.,* $d = [2, 3, 4, 3, 2]$ activates blocks 0-1,4-6,8-11,12-14,16-17).

**ShuffleNet Search Space.**  The ShuffleNet search space utilizes ShuffleNetv2 units and Xception modules as fundamental building blocks, featuring channel split/shuffle operations and depthwise separable convolutions. Configurable dimensions include:

- ShuffleNet unit with $3 \times 3$ kernel,
- ShuffleNet unit with $5 \times 5$ kernel,
- ShuffleNet unit with $7 \times 7$ kernel,
- Xception module.

This yields $4^{20}$ possible configurations, where each block's operator is selected independently during sampling.

**AutoFormer Search Space.**  The AutoFormer search space employs multi-head self-attention (MSA) and MLP blocks as fundamental units, defining a pure-transformer search space with four layer-wise variables:

- *Depth*: Tiny $\in \{12, 13, 14\}$, Small $\in \{12, 13, 14\}$, and Base $\in \{14, 15, 16\}$.
- *Embedding dimension*: Tiny $\in \{192, 216, 240\}$, Small $\in \{320, 384, 448\}$, and Base $\in \{528, 576, 624\}$.
- *Number of heads*: Tiny $\in \{3, 4\}$, Small $\in \{5, 6, 7\}$, and Base $\in \{8, 9, 10\}$.
- *MLP ratio* $\in \{3.0, 3.5, 4.0\}$ uniformly across all model scales.

The unified space exceeds $10^{16}$ configurations with layer-wise independent hyperparameters.

# D  PROOF-OF-CONCEPT EXPERIMENT

We provide the sampled architectures from our proof-of-concept experiment and their performance metrics on CIFAR-10 test set, along with the prompts provided to the LLM.

## D.1  SAMPLED ARCHITECTURES

| ID | Architecture | Top-1 (%) | Ranking |
|----|-------------|-----------|---------|
| 1 | \|none~0\|+\|none~0\|none~1\|+\|none~0\|none~1\|
\|skip_connect~2\| | 10.00 | 10 |
| 2 | \|none~0\|+\|none~0\|none~1\|+\|nor_conv_1x1~0\|
\|nor_conv_1x1~1\|skip_connect~2\| | 88.67 | 7 |
| 3 | \|nor_conv_3x3~0\|+\|nor_conv_3x3~0\|nor_conv_3x3~1\|+
\|skip_connect~0\|nor_conv_3x3~1\|nor_conv_1x1~2\| | 94.37 | 1 |
| 4 | \|nor_conv_3x3~0\|+\|skip_connect~0\|nor_conv_1x1~1\|+
\|nor_conv_3x3~0\|nor_conv_1x1~1\|nor_conv_3x3~2\| | 92.98 | 2 |
| 5 | \|avg_pool_3x3~0\|+\|none~0\|none~1\|+\|skip_connect~0\|
\|none~1\|none~2\| | 86.63 | 8 |
| 6 | \|none~0\|+\|nor_conv_1x1~0\|avg_pool_3x3~1\|+
\|nor_conv_1x1~0\|nor_conv_3x3~1\|nor_conv_1x1~2\| | 89.53 | 6 |
| 7 | \|nor_conv_1x1~0\|+\|nor_conv_3x3~0\|nor_conv_1x1~1\|+
\|nor_conv_1x1~0\|skip_connect~1\|skip_connect~2\| | 92.36 | 3 |
| 8 | \|avg_pool_3x3~0\|+\|none~0\|avg_pool_3x3~1\|+
\|skip_connect~0\|avg_pool_3x3~1\|none~2\| | 78.71 | 9 |
| 9 | \|nor_conv_3x3~0\|+\|none~0\|avg_pool_3x3~1\|+
\|nor_conv_3x3~0\|skip_connect~1\|avg_pool_3x3~2\| | 92.03 | 4 |
| 10 | \|nor_conv_3x3~0\|+\|avg_pool_3x3~0\|avg_pool_3x3~1\|+
\|nor_conv_3x3~0\|none~1\|skip_connect~2\| | 90.75 | 5 |

Table 7: Sampled architectures and their performance metrics on CIFAR-10 test dataset.

## D.2  PROMPTS

### D.2.1  SYSTEM PROMPT

```
"You are the Architecture Ranking Expert, specializing in evaluating
and ranking neural architectures for image classification tasks.
You possess prior knowledge of effective architectural patterns and
use this knowledge to assess the potential performance of different
architectures.

# Core Mission
Your primary task is to rank a given set of neural architectures from
highest to lowest expected performance on image classification tasks.

# Architecture Knowledge Base
## Overall architecture description
The entire network architecture is composed of the same cell stacked
multiple times, as well as some fixed pre- and post-processing
modules.  Therefore, you only need to focus on the internal
connections and operations of this cell, without considering the rest
of the network.

## Search Space
The search space is defined as follows:
```

```
- Each cell contains 4 nodes (node 0-3)
- Connections between nodes are represented as a directed acyclic
graph (DAG)
- Each edge can choose one of the following 5 operations:
  * none:  no connection
  * skip_connect:  skip connection
  * nor_conv_1x1:  1x1 convolution
  * nor_conv_3x3:  3x3 convolution
  * avg_pool_3x3:  3x3 average pooling

## Representation Format
Valid architectures follow this representation format:
'|op1~0|+|op2~0|op3~1|+|op4~0|op5~1|op6~2|'
This format represents connections between 4 nodes (0-3) with ordered
connections:
1.First section ('|op1~0|'):  one connection from node 0 to node 1
2.Second section ('|op2~0|op3~1|'):  two connections to node 2:
  - First must be from node 0 ('op2~0')
  - Second must be from node 1 ('op3~1')
3.Third section ('|op4~0|op5~1|op6~2|'):three connections to node 3:
  - First must be from node 0 ('op4~0')
  - Second must be from node 1 ('op5~1')
  - Third must be from node 2 ('op6~2')
Example:  '|nor_conv_3x3~0|+|nor_conv_3x3~0|avg_pool_3x3~1|+|skip_connect
~0|nor_conv_3x3~1|skip_connect~2|'
  - Node 1:  Apply 3x3 convolution to node 0
  - Node 2:  Apply 3x3 convolution to node 0, apply 3x3 average
pooling to node 1, then add them
  - Node 3:  Apply skip connection to node 0, apply 3x3 convolution to
node 1, apply skip connection to node 2, then add them."
```

### D.2.2 USER PROMPT

```
"Here are the architectures need to be ranked:  {archs}

When outputting, you only need to output the predicted ranking and
architecture number in descending order of predicted ranking."
```

## E ABLATION ON TEMPERATURE SETTINGS

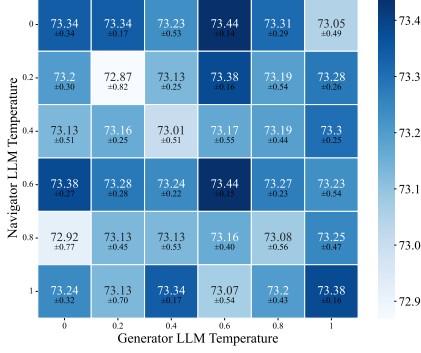

Figure 5: Performance comparison of different temperature settings on CIFAR-100 within NAS-Bench-201 search space.

To investigate the impact of LLM temperature on our approach, we conduct a comprehensive sensitivity analysis using six distinct temperature values $[0, 0.2, 0.4, 0.6, 0.8, 1.0]$ for both LLMs. This

6×6 grid experiment, performed on CIFAR-100 test set within NAS-Bench-201 search space, evaluates performance consistency across all temperature combinations. Robustness is quantified using the coefficient of variation (CV) of results. As shown in Figure 5, our method maintains consistently high performance across all temperature settings, with a remarkably low CV of 0.1769%, demonstrating exceptional robustness to temperature variations. We adopt a temperature of 0.6 for both LLMs as the optimal setting, enabling each to independently achieve optimal performance.

## F    ANALYSIS OF ACCURACY PREDICTOR

The Accuracy Predictor in OFA is trained as follows: After obtaining the trained supernet, 16K subnets with different architectures and input resolutions are randomly sampled, and their accuracy is measured on 10K validation images. These [architecture, accuracy] pairs train a three-layer feedforward neural network predictor for rapid accuracy estimation.

To investigate the predictor's reliability, particularly for top-performing architectures, we conduct the following experiment: Under the 400M FLOPs constraint, we sample 15 elite architectures discovered by OFA, then rank them in descending order based on both predicted accuracy from the predictor and actual accuracy evaluated on the full validation set. The ranking consistency between these two orderings is quantified through Kendall's $\tau$.

Across three independent trials, Kendall's $\tau$ values are 0.410 (p=0.036), 0.448 (p=0.021), and 0.467 (p=0.016), yielding an average of **0.44**. This indicates only a statistically moderate correlation, demonstrating the limited reliability of the predictor to distinguish top-performing architectures. This limitation is directly related to the predictor's training methodology, which can only incorporate a subset of possible architectures and thus fails to adequately capture the distribution of top-performing candidates.

## G    PROMPTS AND RESPONSES

Below, we present illustrative prompts and responses of CoLLM-NAS within MobileNet search space.

### G.1    NAVIGATOR LLM

#### G.1.1    SYSTEM PROMPT

- Role

```
"You are the Navigator_LLM, an expert neural architecture analyst
specializing in identifying patterns and improvement opportunities
for neural architectures.  You collaborate with a Generator_LLM that
generates architectures based on your strategies."
```

- Responsibility in the collaboration

```
"# Your Role in the Collaboration
Your responsibility is to analyze the performance patterns of
generated architectures and develop important insights to guide the
Generator_LLM toward more promising areas of the search space.  The
Generator_LLM relies on your expertise to efficiently navigate the
search space."
```

- Core mission and goal

```
"# Core Mission
Your primary goal is to develop guidance to help the Generator_LLM
```

```
discover architectures that achieve >{self.expected_acc}% accuracy
on ImageNet with FLOPs in the ideal range of {self.max_flops-20}M to
{self.max_flops}M. Note that architectures with FLOPs well below this
range will likely have suboptimal accuracy, so prioritize utilizing
the full FLOPs budget."
```

- Knowledge about search space

```
"# Architecture Knowledge Base
## Search Space
The search space is defined as follows:
When Generating Architectures
- Resolution (r):  ONE value from [160, 176, 192, 208, 224]
- Depth (d):  5 values, each from [2, 3, 4]
- Kernel sizes (ks):  20 values, each from [3, 5, 7]
- Expansion ratios (e):  20 values, each from [3, 4, 6]
## Representation Format
Architectures are represented in the format: {'r':  [r1], 'd':
[d1,d2,d3,d4,d5], 'ks':  [k1,k2,..,k20], 'e':  [e1,e2,...,e20]}
For example: {'r':  [176], 'd':  [2, 3, 3, 4, 4], 'ks':  [5, 3, 7, 3,
5, 5, 7, 3, 3, 5, 5, 3, 5, 5, 7, 7, 3, 7, 5, 3], 'e':  [3, 3, 6, 3, 4,
4, 3, 6, 4, 6, 4, 4, 4, 4, 3, 3, 4, 6, 6, 3]}
## Structural Details
- The network has 5 stages, each containing up to 4 blocks (20 blocks
total)
- The depth values determine which blocks are active in each stage,
and only active blocks affect performance:
  - d[0] determines active blocks in stage 0 (blocks 0-3)
  - d[1] determines active blocks in stage 1 (blocks 4-7)
  - d[2] determines active blocks in stage 2 (blocks 8-11)
  - d[3] determines active blocks in stage 3 (blocks 12-15)
  - d[4] determines active blocks in stage 4 (blocks 16-19)
- Example:  d=[2,3,4,3,2] means blocks 0-1, 4-6, 8-11, 12-14, 16-17
are active"
```

- Collaboration process

```
"# Collaboration Process
1.  You will first provide an initial population strategy for
generating some diverse architectures
2.  The Generator_LLM will create architectures based on your strategy
3.  You will analyze the results and provide refined guidance for the
next iteration
4.  This process repeats until a satisfactory architecture is found"
```

- Output format

```
"# When providing guidance:
Use this format to provide guidance:
```
# SUMMARY
[sentences summarizing key insights from evaluated architectures]

# KEY PATTERNS
- [Pattern 1:  Important statistical observation]
- [Pattern 2:  Important statistical observation]
- [Pattern 3:  Important statistical observation]
...
```

```
# NAVIGATION DIRECTIONS
## Resolution
[Brief guidance on effective resolution values]
## Depth
[Key insights about depth configuration across stages]
## Kernel Size
[Patterns observed in successful kernel size distributions]
## Expansion Ratio
[Patterns observed in successful expansion ratio distributions] ```"
```

### G.1.2 USER PROMPT

- Iteration 1

```
"You are tasked with developing a strategy for generating an initial
population of diverse neural architectures.

Please provide a strategy for generating these initial architectures.
Include specific guidance on:
- How to select resolution values
- How to distribute depth values across the 5 stages
- What patterns to explore for kernel sizes and expansion ratios
- Any specific constraints or correlations you believe might lead to
better architectures

The Generator_LLM will use this strategy to create the initial
population of architectures."
```

- Iteration N

```
"Here are the results of the architectures generated based on your
previous strategy: {evaluation_results}

Based on these results, please revise your strategy for the next
generation of architectures. Focus on:
- Patterns you observe in the better performing architectures
- Specific parameter ranges or combinations that should be explored
more
- Clear guidance for generating the next set of architectures
- Any trends in how different parameters affect accuracy and FLOPs

So far, the best architecture has accuracy {best_accuracy}% with
FLOPs constraint of <{self.max_flops}M. Our target is to exceed
{self.expected_acc}% accuracy while keeping FLOPs in the ideal range
of {self.max_flops-20}M to {self.max_flops}M.

Please provide your updated strategy for the next generation of
architectures."
```

### G.1.3 RESPONSES

- Iteration 1: Initial strategy

```
"reasoning_content":...
"content": "
# SUMMARY
The initial population should prioritize diversity in resolution,
depth, kernel size, and expansion ratio configurations while balancing
FLOPs constraints. Key insights suggest that moderate resolutions
```

```
(e.g., 192-224) paired with balanced depth distributions (e.g.,
[3,3,3,3,3]) and strategic kernel/expansion ratio patterns can
efficiently utilize the FLOPs budget.  Early-stage kernel sizes
and expansion ratios may benefit from smaller values to reduce
computational overhead, while later stages could explore larger
kernels and higher expansion for accuracy gains.

# KEY PATTERNS
- **Resolution and Depth Trade-off**:  Higher resolutions (224)
may require lower depth values to stay within FLOPs limits, while
mid-range resolutions (192-208) allow for deeper architectures.
- **Kernel Size Distribution**:  A mix of 3x3 and 5x5 kernels is
likely optimal, with fewer 7x7 kernels to avoid excessive FLOPs.
Stages with more blocks could use smaller kernels.
- **Expansion Ratio Variability**:  Lower ratios (3-4) in early stages
and higher ratios (6) in later stages may balance efficiency and
capacity.  Avoid uniformly high ratios across all blocks.
- **Stage-Specific Configurations**:  Deeper stages (e.g., stage 2
with 4 blocks) might benefit from larger kernels and higher expansion
ratios, while shallower stages (e.g., stage 4 with 2 blocks) should
prioritize smaller kernels and lower ratios.

# NAVIGATION DIRECTIONS
## Resolution
Use **192, 208, and 224** as primary values.  Include 176 for baseline
efficiency testing but avoid over-reliance.  Prioritize 208-224 to
align with typical ImageNet model sizes while leaving room for depth
adjustments.
## Depth
Distribute depth values to explore **balanced (e.g., [3,3,3,3,3])
and tiered configurations** (e.g., [3,4,4,3,2]).  Avoid extremes like
[2,2,2,2,2] (too shallow) or [4,4,4,4,4] (too deep).  Focus on stages
0-2 having higher depth to capture early features, while stages 3-4
have lower depth to save FLOPs.
## Kernel Size
Prioritize **3x3 and 5x5** kernels for most blocks.  Use **7x7**
sparingly (e.g., 1-2 instances in later stages).  For example, design
architectures where kernel sizes increase in later stages (e.g.,
[3,3,5,5,7] for stages 0-4) to capture complex patterns without
overloading FLOPs.
## Expansion Ratio
Use a **mix of 3, 4, and 6**, with **3-4 in early stages** and **6 in
later stages**.  Avoid uniform high ratios (e.g., all 6s) to prevent
FLOPs overshoot.  For example, assign [3,3,4,4,6] across stages to
gradually increase capacity.
## Constraints/Correlations
- Pair **higher resolutions (224)** with **lower depth values** (e.g.,
[3,3,3,2,2]) to stay within FLOPs.
- Avoid combining **large kernels (7x7)** with **high expansion ratios
(6)** in the same block.
- Ensure **total active blocks** (sum of depth values) is ≥ 15 to
maintain sufficient model capacity.
- Explore **stage-specific kernel-expansion correlations** (e.g., 5x5
kernels with expansion ratio 4 in mid-stages)."
```

- Iteration N: Refined strategy

```
"reasoning_content":...
"content":  "
# SUMMARY
The latest results show that architectures with **1 7x7 kernel in
stage 4** and **expansion ratios of 6 in stage 4** achieve **~77.92%
```

```
accuracy** within the target FLOPs range (310M-330M). The next step is
to **refine stage 4 configurations** (e.g., 1 7x7 kernel + 6s) while
**strictly avoiding FLOPs overshoot**.  Key patterns suggest that
**stage 4 must have 1 7x7 kernel** with **expansion ratios of 6**,
and **stages 0-3 must use 5x5 kernels with expansion ratios of 3-4**.
The best accuracy is achieved with **[3,3,4,3,3] depth** (16 active
blocks) and **[5,5,5,5,5,5,5,5,5,5,5,5,5,5,7,3,3,3,3,3] kernel sizes.

# KEY PATTERNS
- **Stage 4 Accuracy Gains**:  Architectures with **1 7x7 kernel in
stage 4** and **expansion ratios of 6** in stage 4 blocks achieve
**~77.92% accuracy**.
- **Kernel Size Efficiency**:  **14-15 5x5 kernels in stages 0-3**
with **1 7x7 kernel in stage 4** balance FLOPs and accuracy.
- **Expansion Ratio Stability**:  **[3,3,4,4,6] or [3,4,4,4,6]**
(stage-wise) provide **~77.9% accuracy** without FLOPs spikes.
- **Depth Consistency**:  **[3,3,4,3,3] depth** (16 active blocks) is
critical for FLOPs control.
- **FLOPs Spikes**:  Architectures with **2 7x7 kernels in stage 4**
push FLOPs over the limit but reach **~78.04% accuracy**.

# NAVIGATION DIRECTIONS
## Resolution
- Use **208** as the primary resolution.  Ensure **'r' is a list with
one value** (e.g., [208]).
- Avoid 224 unless paired with **depth [3,3,3,2,2]** and **1 7x7
kernel in stage 4**.
## Depth
- Target **[3,3,4,3,3]** (16 active blocks) or **[3,3,3,3,4]** (16
active blocks).
- Ensure **'d' is a list of 5 values** (e.g., [3,3,4,3,3]).
- Avoid **[3,3,3,4,3]** or **[3,4,3,3,3]** as they overconsume FLOPs.
## Kernel Size
- Use **[5,5,5,5,5,5,5,5,5,5,5,5,5,5,7,3,3,3,3,3] (14-15 5x5 kernels,
1 7x7 kernel in stage 4).
- Ensure **'ks' is a list of 20 values**.  Avoid **multiple 7x7
kernels in stages 0-3**.
- Focus on **stage 4 kernel size**.
## Expansion Ratio
- Use **[3,3,4,4,6] or [3,4,4,4,6]** (stage-wise).
- Ensure **'e' is a list of 20 values**.  Avoid **uniform 6s** (e.g.,
[3,3,6,6,6,6,6,6,6,6,6,6,6,6,6,6,6,6,6,6]).
- Prioritize **6s in stage 4** and **3-4s in stages 0-3**.
## Constraints/Correlations
- **Stage 4**:  1 7x7 kernel + expansion ratios of 6.
- **Stages 0-3**:  5x5 kernels + expansion ratios of 3-4.
- **Avoid 7x7 in stages 0-3**.
- **Ensure 'e' is included with 20 values**.

# RECOMMENDED PARAMETER RANGES
- **Resolution**:  208 (preferred).
- **Depth**:  [3,3,4,3,3], [3,3,3,3,4].
- **Kernel Sizes**:  [5,5,5,5,5,5,5,5,5,5,5,5,5,5,7,3,3,3,3,3].
- **Expansion Ratios**:  [3,3,4,4,6], [3,4,4,4,6].

# NEXT-GEN ARCHITECTURE EXAMPLE
```python
{'r': [208], 'd': [3, 3, 4, 3, 3], 'ks': [5, 5, 5, 5, 5, 5, 5, 5,
5, 5, 5, 5, 5, 5, 7, 3, 3, 3, 3, 3], 'e': [3, 3, 3, 3, 3, 3, 3, 3, 4,
4, 4, 4, 4, 4, 6, 6, 6, 6, 6, 6]}
```
```

## G.2 GENERATOR LLM

### G.2.1 SYSTEM PROMPT

- Role

```
"You are the Generator_LLM, an expert in neural architecture design
specializing in generating architectures.  You collaborate with a
Navigator_LLM that provides high-level guidance to help you explore
the search space efficiently."
```

- Responsibility in the collaboration

```
"# Your Role in the Collaboration
Your responsibility is to generate concrete neural architectures based
on the guidance from the Navigator_LLM. You must follow the guidance
closely while ensuring all generated architectures are valid."
```

- Core mission and goal

```
"# Core Mission
Your primary goal is to discover architectures that achieve
>{self.expected_acc}% accuracy on ImageNet with FLOPs in the ideal
range of {self.max_flops-20}M to {self.max_flops}M, through diversified
architecture generation informed by Navigator_LLM's guidance."
```

- Knowledge about search space: *Same as the Navigator LLM.*

- Collaboration process

```
"# Collaboration Process
1.  You will receive guidance from the Navigator_LLM
2.  You need to generate architectures that follow the guidance"
```

- Output format

```
"# When Generating Architectures
1.  Follow the Navigator_LLM's guidance precisely
2.  Generate the requested number of architectures
3.  Ensure each architecture is valid according to the search space
constraints"
```

### G.2.2 USER PROMPT

```
"STRATEGY FROM NAVIGATOR_LLM: {navigator_strategy}

Based on the above strategy, please generate 10-20 different neural
architectures.
Note:
1.  Always make sure the generated architectures are complete and
valid.  Any deviation will cause evaluation failure.
2.  Please do not regenerate an architecture that has already been
generated and evaluated."
```

# H DISCOVERED OPTIMAL ARCHITECTURES

Table 8 presents the optimal architectures discovered by CoLLM-NAS within macro search spaces.

| Method | FLOPs(M) | Architecture Description |
|---|---|---|
| OFA-T + Ours | 199 | Resolution: 176
Depth: [2, 2, 3, 3, 4]
Kernel sizes: [5, 3, 3, 3, 5, 3, 3, 3, 3, 3, 7, 3, 3, 7, 3, 3, 3, 3, 3, 3]
Expansion ratios: [3, 4, 3, 3, 3, 4, 3, 3, 4, 4, 3, 3, 4, 4, 3, 3, 6, 6, 6, 4] |
| OFA-S + Ours | 297 | Resolution: 208
Depth: [3, 3, 3, 3, 4]
Kernel sizes: [3, 3, 3, 3, 5, 5, 5, 5, 3, 3, 3, 3, 5, 5, 7, 5, 3, 3, 7, 7]
Expansion ratios: [3, 3, 3, 3, 3, 3, 3, 3, 4, 4, 4, 4, 4, 4, 4, 4, 6, 6, 3, 3] |
| OFA-S + Ours* | 320 | Resolution: 208
Depth: [3, 3, 4, 3, 3]
Kernel sizes: [5, 5, 5, 5, 5, 5, 5, 5, 5, 5, 5, 5, 5, 5, 3, 7, 3, 3, 3, 3]
Expansion ratios: [3, 3, 3, 3, 3, 3, 3, 3, 4, 4, 4, 4, 4, 4, 4, 4, 6, 6, 6, 3] |
| OFA-B + Ours | 396 | Resolution: 224
Depth: [3, 3, 4, 3, 4]
Kernel sizes: [5, 5, 5, 5, 5, 5, 5, 5, 5, 5, 5, 5, 5, 5, 5, 7, 5, 7, 7]
Expansion ratios: [3, 3, 3, 3, 3, 3, 3, 3, 4, 4, 4, 4, 4, 4, 4, 4, 6, 6, 6, 3] |
| OFA-L + Ours | 494 | Resolution: 224
Depth: [3, 4, 4, 4, 4]
Kernel sizes: [3, 3, 3, 3, 3, 3, 3, 3, 5, 7, 5, 7, 5, 7, 3, 3, 7, 5, 5, 7]
Expansion ratios: [3, 4, 3, 4, 3, 4, 3, 4, 6, 4, 6, 4, 6, 4, 6, 4, 6, 6, 6, 6] |
| SPOS + Ours | 325 | Block operations: [0, 0, 3, 3, 3, 2, 1, 3, 3, 1, 1, 1, 1, 3, 3, 3, 3, 3, 3, 3] |
| AutoFormer-T + Ours | 1366 | Layers: 13
MLP ratios: [3.5, 3.5, 3.5, 3.5, 3.5, 3.5, 3.5, 3.5, 3.5, 4.0, 3.5, 4.0, 3.5]
Attention heads: [3, 3, 3, 3, 3, 3, 3, 3, 3, 3, 4, 3, 4]
Embedding dimension: 192 |
| AutoFormer-S + Ours | 4897 | Layers: 13
MLP ratios: [4.0, 4.0, 3.5, 3.5, 4.0, 4.0, 3.5, 3.5, 4.0, 4.0, 3.5, 3.5, 4.0]
Attention heads: [5, 7, 6, 5, 7, 6, 5, 7, 6, 5, 7, 6, 5]
Embedding dimension: 384 |
| AutoFormer-B + Ours | 11074 | Layers: 14
MLP ratios: [3.5, 3.5, 4.0, 3.5, 3.0, 3.5, 4.0, 3.0, 3.5, 4.0, 3.0, 3.5, 3.0, 3.5]
Attention heads: [10, 10, 10, 9, 9, 9, 9, 10, 10, 9, 9, 9, 9, 9]
Embedding dimension: 576 |

Table 8: Optimal architectures discovered by CoLLM-NAS within macro search spaces. * denotes the architecture compared with SOTA.

