# OpenReview forum: "CoLLM-NAS: Collaborative Large Language Models for Efficient Knowledge-Guided Neural Architecture Search"
_ICLR.cc/2026/Conference — ICLR 2026 Conference Withdrawn Submission_

### Official Review · Reviewer_eeBN · 2025-10-29

**Soundness:** 2
**Presentation:** 2
**Contribution:** 2
**Rating:** 4
**Confidence:** 4

**Summary:**

This paper proposes a collaborative LLM-based NAS framework to discover high-performing architectures. The collaborative framework contains stateful Navigator LLM, stateless Generator LLM, and a coordinator module. The whole framework is built on the two-stage NAS and employs the pretrained supernet for evaluating the generated architectures. Experimental results on macro and micro search spaces show the effectiveness of the proposed method.

**Strengths:**

1.	The paper is well-written and easy to understand.

2.	The proposed framework leverages both LLM’s inherent knowledge and progressive insights learned from historical trajectory to guide the NAS strategy.

3.	The experimental results demonstrate the proposed collaborative LLM-based NAS framework.

**Weaknesses:**

1.	The novelty is limited. The framework divides the LLM NAS agent into three roles, e.g., Navigator, Generator, and Coordinator in a straightforward manner. However, the core idea is also leveraging the LLM to generate the architecture. There is no significant innovation.

2.	The paper lacks some theoretical analysis. It is not clear why the collaborative farmwork containing Navigator and Generator works.

3.	The coordinator module checks the legality of the generated architecture. It should be better to calculate the proportion of illegal architectures in the overall search process. And, the effectiveness of Generator for generate legal architectures could be further evaluated.

4.	In Table 1 and Table 2, there lacks statistical performance. Since the output of LLM is probabilistic, multiple runs should be performed and statistical performance should be computed.

5.	The proposed framework relies on a pre-trained supernet. Thus, the search performance is affected by the quality of the pre-trained supernet. For example, in Table 2, the authors only show the results based on OFA. How is the performance using the pre-trained supernets of other two-stage NAS methods?

6.	In Table 1, What do ‘T’, ‘S’, ‘B’, ’L’ mean? The authors should introduce the meaning.

7.	It is suggested to add the experimental evaluation about the token cost during the whole search process.

**Questions:**

see Weaknesses.

---

### Official Review · Reviewer_ShpM · 2025-10-31

**Soundness:** 2
**Presentation:** 3
**Contribution:** 2
**Rating:** 2
**Confidence:** 4

**Summary:**

CoLLM-NAS replaces the NAS search phase with an LLM-guided process. A Navigator LLM maintains objectives, and past feedback to suggest search directions, and a stateless Generator LLM turns those directions into valid architectures within the defined space. A coordinator verifies legality, and scores candidates using a pretrained supernet with weight inheritance to avoid retraining. This setup aims to achieve efficient exploration without traditional evolutionary or reinforcement learning search.

**Strengths:**

1. The devised proof-of-concept experiment effectively demonstrates that large language models exhibit non-trivial comprehension of neural architecture performance patterns.

2. The authors demonstrated the effectiveness of CoLLM-NAS across three macro search spaces—MobileNet, ShuffleNet, and AutoFormer—and one micro search space, NAS-Bench-201, although the observed performance improvements are relatively marginal.

3. The presentation of the algorithm is clear and easy to follow.

**Weaknesses:**

1. The proposed method appears to lack novelty. The use of a pretrained supernet for efficient evaluation has already been explored in prior NAS work, and the proposed collaboration mechanism seems to amount to a simple adoption of multiple LLMs, which may not constitute a strong contribution on its own.

2. As shown in Table 1, the performance improvements achieved by adding the proposed method to OFA, SPOS, and AutoFormer are quite marginal compared to their respective baselines. While it can be argued that the method is efficient under a limited architecture budget, it would strengthen the paper to demonstrate how the discovered architectures perform when the search budget is increased, as this would further validate the effectiveness of the proposed approach.

3. The motivation for separating the Navigator LLMs and Generator LLM is not clearly justified. It would be helpful to include a comparison with a single LLM that both guides the search process and generates architectures (e.g., agentic LLMs for NAS). Additionally, cases involving illegal architecture configurations might be more effectively addressed through constrained decoding strategies.

**Questions:**

1. How were the ten neural architectures in the proof-of-concept experiment sampled?

2. Could you explain in more detail what specific role the generator LLMs play, and why they are necessary as a separate component?

---

### Official Review · Reviewer_7mp5 · 2025-11-02

**Soundness:** 3
**Presentation:** 2
**Contribution:** 2
**Rating:** 4
**Confidence:** 4

**Summary:**

This paper introduces Large Language Models (LLMs) to two-stage NAS and further proposes the CoLLm-NAS. CoLLm-NAS mainly consists of three key modules: a navigator LLM, a generator LLM and the Coordinator. The Navigator LLM keeps the persistent memory across iterations and controls the search policy from exploration to exploitation. The generator LLM  follows  the guidance  Navigator LLM to generate new architectures. Finally, the Coordinator maintains the overall workflow . CoLLm-NAS is avaluated on macro search spaces: MobileNet, ShuffleNet, and AutoFormer, and one micro cell-based search space: NAS-Bench-201. According to the results on the experiment section,  CoLLm-NAS  achieves SOTA results across the above search spaces and various supernet training strategies.

**Strengths:**

1. This paper proposes a novel collaborative LLM-NAS framework (CoLLM-NAS), which for the first time integrates two complementary LLMs (Navigator and Generator) with two-stage NAS, demonstrating clear innovation. Designs three core components (Navigator LLM, Generator LLM, Coordinator) and clearly defines their roles and collaborative mechanisms within the search process, resulting in a well-structured framework.
2. Comprehensive and experiments：conducts extensive experiments across multiple mainstream search spaces (MobileNet, ShuffleNet, AutoFormer, NAS-Bench-201), validating the method's generality and superiority. Achieves state-of-the-art (SOTA) results on both ImageNet and NAS-Bench-201, significantly outperforming conventional search algorithms (e.g., EA, RL) and existing LLM-based NAS methods.
3. Search efficiency is markedly improved, with search costs reduced by 3–10 times, highlighting the method's potential for practical application.

**Weaknesses:**

1. In Section 3.1, which validates the LLM's architecture comprehension, the process of how the LLM predicts the performance of model architectures and ranks them according to these predictions is not clearly explained. Furthermore, this validation experiment only involves predicting and ranking 10 neural architectures over multiple trials. Would considering a ranking of the entire NAS-Bench-201 search space provide a more comprehensive verification of the LLM's understanding of model architectures?
2. While the results from both the macro search spaces and the micro cell-based search space demonstrate that CoLLM-NAS indeed achieves state-of-the-art (SOTA) performance, the improvements over the baselines are relatively marginal. Further enhancements to the algorithm's effectiveness may be necessary.
3. The paper lacks theoretical analysis or explainability research on why LLMs can effectively guide NAS. Although a preliminary architecture ranking experiment is provided, there is insufficient in-depth exploration into the mechanisms of how LLMs comprehend architectures and formulate strategies. Furthermore, the potential for enhancing search performance by incorporating additional domain-specific knowledge of NAS remains unexamined.

**Questions:**

My primary concern is that the authors posit that leveraging the LLM's capacity for architectural comprehension to establish a global memory capability is beneficial for accelerating the search efficiency and discovering optimal solutions. Although the authors discuss in Section 4.3 why CoLLM-NAS achieves superior results, the underlying principles remain conceptually unclear.

---

### Note · Authors · 2025-11-14

I have read and agree with the venue's withdrawal policy on behalf of myself and my co-authors.